# Impact of Coordination, Psychological Safety, and Job Security on Employees' Performance: The Moderating Role of Coercive Pressure

**Yong Ming Wang [1], Waqar Ahmad [1,\*], Muhammad Arshad [2,\*], Hong Li Yin [3], Bilal Ahmed [4] and Zulfiqar Ali [1]**

1    Faculty of Management and Economics, Kunming University of Science and Technology, Kunming 650500, China; Ym_w@kmust.edu.cn (Y.M.W.); zulf_hunzai@yahoo.com (Z.A.)
2    School of Business Management, Yunnan University of Finance and Economics, Kunming 650500, China
3    School of Information Science and Technology, Yunnan Normal University, Kunming 650093, China; hongli_yin@163.com
4    Institute of International Relations, South and Southeast Asian Institute, Yunnan University, Kunming 650500, China; bahishtbilal@gmail.com
\*    Correspondence: waqar_ah@stu.kust.edu.cn (W.A.); arshadtanoli8712@gmail.com (M.A.)

**Abstract:** Based on institutional theory, the current study examines the relationship of coordination, job security, psychological safety, and coercive pressure with employees' performance. Further, coercive pressure is treated as a boundary condition between coordination, job security, and psychological safety with employees' performance. A survey method was used to collect data from 235 faculty members of higher education institutions (HEIs) in Pakistan. Study results show that there is a positive and significant relationship between job security and coordination with employees' performance. The results also reveal that coercive pressure moderates job security, coordination, and psychological safety with employees' performance. Implications for organizations and HEIs administration are discussed.

**Keywords:** coordination; employees' performance; coordination; job security; psychological safety; coercive forces; higher educational institutions of Pakistan

## 1. Introduction

Employees are the most important element for an organization; indeed, they are the most important asset of an organization. Their performance in an organization collectively affects the organization's performance [1]. So, performance is like behavior that improves the capacity of the employees [2]. Further, employee performance for the organization is a motivating factor to strengthen the organization's performance as a whole [3]. Employees' performance is relevant to the behavior and action of the employees at the workplace, which is a link to the organization's goals [4]. So, the employees' (faculty members) performance plays a significant role in all types of organizations, including higher education [5].

Higher education has a primary role in enhancing the nation's knowledge economy [6]. Any country's knowledge indicators predict the development of a society [7]. It is imperative for the education sector's higher authority to underpin those problems, which are associated with the HEIs and their employees. HEIs' employees' performance has unique importance for the knowledge economy of a country, which is still not given much attention from the actors for the survival of HEIs' quality and research [8].

In past studies, researchers explored the relationship of psychological safety with an individual's work performance [9,10], psychological safety with a mediating variable workplace bullying and harassment [11], psychological safety and creative performance [12,13], coordination and employees' performance [14,15]. All these previous researchers' work explains the importance of the relationship between psychological safety, coordination, and job security with employee performance. Therefore, coordination, psychological safety,

and job security play a significant role in enhancing employees' performance in the organization [16,17]. So, it is clear from the previous research that individual relationships of these variables have been explored with different dependent variables, but together, these three variables coordination, psychological safety, and employees' performance are unexplored so far.

According to [18], Institutional Theory explains there are three institutional pressures—those which have an impact on the organization's and its employees' performance. These institutional pressures are coercive, mimetic, and normative [18,19]. From the three institutional pressures, coercive pressure is more important because it relates to rules, regulations, punishments, and sanctions by government regulations and other shareholders that influence employees' performance [20]. This pressure changes institutional practices, which ultimately affect the organization and its employees. Streams of studies explain that coercive pressure has a strong role in the long life and development of an organization and its employees [21] and compels an organization to improve its performance [22]. Furthermore, coercive pressure has importance for educational institutions that enforce an organization to meet leading organization standards [23,24]. However, the authors of [25] explain that coercive pressure is important for organizational fit. So, it is imperative to address this issue on a priority basis. Moreover, it is assumed that coercive pressure may moderate the relationship of coordination, job security, and psychological safety with employee performance (Figure 1 shows study model). Streams of research explain that no such studies were carried out to fill the theoretical void between moderating variable (coercive pressure) coordination, job security, psychological safety, and employees' performance in Pakistan's context [26].

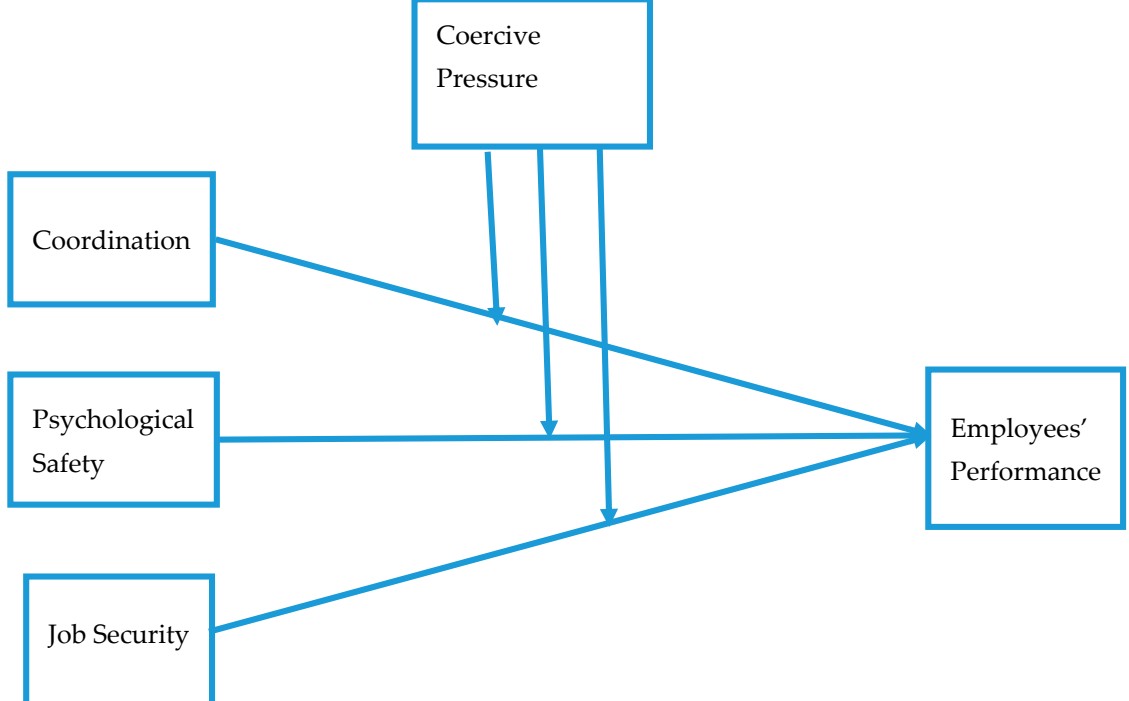

**Figure 1.** Proposed research frame.

The current study will attempt to answer these questions: (a) Does coordination, psychological safety, job security have a positive relationship with employees' performance? (b) Does coercive pressure moderate the relationship between coordination and employee performance? (c) Does coercive pressure moderate the relationship between job security and employee performance (EP)? (d) Does coercive pressure strengthen the relationship between psychological safety and an employee's performance?

This study tries to fill the theoretical void, which is so far rarely discussed by researchers. This study framework contributes to the study in several ways in the literature of employee performance. Firstly, it investigates the relationship between coordination, job security, psychological safety, and employee performance. Secondly, coercive pressure plays a role in strengthening the relationship between coordination, job security, psychological safety, and employee performance.

## 2. Theoretical Background and Hypothesis Testing

### 2.1. Employees' Performance

Employees' performance is a vital building block of an organization, a pre-requisite for an organization's success. In other words, employees' performance directly affects an organization's progress and efficiency [27]. According to the study in [28], it explains that employees' performance has a significant and multidimensional role that aims to achieve results that strongly link to an organization's strategic goals. It means that employees' performance measures both efficiency and productivity [16]. Employees' performance has a relationship with the job performance of the employees, which has a final effect on an employee's effort, as modified by his abilities, roles, or task perceptions [29,30] which is integrated with the job satisfaction and commitment of employees with the organization. The relationship between job performance and feedback has been studied extensively in motivational theories, such as social learning theory [31]. According to this theory, employees' performance includes establishing their performance appraisal plans' elements and standards. Employees' performance should be measurable, understandable, verifiable, equitable, and achievable. Streams of research explain that employee performance has importance in educational sectors, illuminating and magnifying its importance [32]. Scholars have described the need for an improved understanding of job security and satisfaction in an organization; this factor should be fulfilled for results [33], although there are various approaches and benefits for an organization to motivate the employees for their better performance. Due to employees' performance, the organization's productivity increased which ultimately affected the organization economically [34].

### 2.2. Coordination and Employees' Performance

Coordination has generally been known as a mechanism to regulate interdependent objects in the form of the tasks, responsibilities, capabilities, and information of different groups that must match both ways and time for an integrated accomplishment of organizational objectives [35]. Coordination has an integrated role in the group of people in an organization to work in a team to tackle issues and minimizes the obstacles for individuals and an organization's goal [36]. It is essential to enhance employees' motivation and employees' voice and management, as well as develop, and define their role and responsibilities [37]. According to the study in [38] "coordination is putting the pieces together, we do not have an organization but only a collection of separate activities". Coordination is the actual relationship when the people in the group work together to achieve a common goal.

Researchers in [39] believe that coordination has a positive and significant role in the development of employees' performance. Further, Mohiuddin [40] and Akhtar et al., [41] have explained that coordination plays a crucial role in an organization's commitment and performance. Further, coordination has significance in the education sector to improve the performance of the faculty members [42]. Therefore, based on the above arguments, it can be analyzed that coordination has importance for the employees' performance in an organization. So, the following hypothesis can be assumed as:

**Hypothesis 1 (H1).** *Coordination has a positive and significant influence on employees' performance.*

### 2.3. Psychological Safety and Employees' Performance

Psychological safety is a sense of showing and employing self-respect without negative consequences [43]. Psychological safety can also be clarified as a psychological climate,

where employees feel safe [10]. The studies reveal that psychological safety is the need for the employees' safety and security on the job, which affects the organization's performance. According to the study in [44], the authors suggested a climate of psychological safety allows people to feel safe while taking risks, setting high standards that require enormous effort, and such a type of interdependent safe environment to perform their job. Further research [44] explains that psychological safety promotes situations where employees can learn and improve their performance safely. It indicated that psychological safety is an essential relational catalyst for employees' learning behaviors in the organizational work setting for their performance. Psychological safety and trust environment play a significant role in fueling intrinsic motivation [44]. Psychological safety helps develop the employees' confidence, and it ultimately strengthens the courage of the employees to take risks and show performance [45,46]. Psychological safety emphasizes an organization's environment where critical or reflective thinking is openly exercised [47]. Psychological safety could be developed by facilitating high-quality relationships of shared goals, knowledge, information, planning, and mutual respect among employees [48]. This explains that psychological safety has a role in changing the attitude and behavior of the employees [49].

Psychological safety for the employees' performance is an essential factor for an organization's survival [50]. It carries great importance for the organization's good environment. For knowledge sharing, the perceptions of psychological safety are vital for employees. Furthermore, from the work in [50], study findings expanded our knowledge that different employees react differently to HR practices within the same organization environment, so an organization needs to pay attention to the psychological safety of employees' performance factors. A psychologically safe environment within the organization has an alignment with its employees and organization performance [10,51].

Furthermore, the researcher indicated that psychological safety within the organization positively links with its stakeholders, ultimately giving appositive moves for the organization employees and its other stakeholders [45]. Therefore, psychological safety has a supporting role in the employees' performance at the workplace. Based on the above discussion, our hypothesis is as follows:

**Hypothesis 2 (H2).** *Psychological safety has a statistically positive effect on the employees' performance.*

### 2.4. Job Security and Employees' Performance

Job security can be defined as security associated with a decline in average tenure and an increase in employment exit rates [52]. Organizations need human resources to accomplish their goals, and also, employees need job security and satisfaction at the workplace [53]. So, this factor has importance for employees. As job security has a significant role, so researchers have worked extensively and defined it differently. As per the study in [54], studies explain that job security is the main factor for accomplishing organizational tasks by securing employees' jobs. Furthermore, Lăzăroiu [55] has explained that job security is an essential factor for an organization's motivation. Job insecurity [56] has diverse and risky effects that affect the employees psychologically and economically and ultimately affect the organization's goals and objectives. Therefore, job security has an integrated role in the organization, primarily for its performance [17]. Suppose the employees observe distributive justice and coordination within the organization regarding their job security and satisfaction [56]. In that case, they perform well and show their involvement within the organization and with the organization's customers. New policies and institutional pressure have importance for the organization that is still suffering to address employees' job security [53]. The issues and problems arise in most organizations where most employees feel insecure and psychologically depressed. So, there needs to be attention to address these issues on a priority basis [57,58]. Job insecurity creates the risk of unemployment for an employee, which usually has negative psychological consequences and affects employees'

performance [59]. Therefore, this issue needs to address the organization's productivity and employees' satisfaction. Based on the above arguments, the following hypothesis proposed:

**Hypothesis 3 (H3).** *Job security has a positive and significant relationship with employees' performance.*

### 2.5. The Moderating Role of Coercive Pressure

Institutional factors play an essential role in employees' performance enhancement [60,61]. The coercive pressures significantly affect an organization because it compels the organization to follow the rules and regulations of higher authority, vital for its survival in a competitive environment [62]. Educational institutions have increasingly been subjected to the stakeholders' enormous pressure and the education department to improve performance [63,64]. As in the context of Europe, the role of the institutional environment has not shown a significant difference in education sector performance [64,65]. The increased attention on performance measures by academics and consultants reflects the increased pressure to enhance organizational performance [66]. The significant relationship between coercive pressure and organizational performance is evident in recent literature [19].

Further, according to the findings of [67] that universities and academic workers are affected by external pressures related to higher education that include government regulations and control of the state (state pressure), the expectations of the professional norms, and collegiality of the academic community (academic pressures)compel the institutions to adapt themselves as according to requirements of the main actors. In the European context, the institutional pressures have a partial role in the performance of the institutions [68]. Institutional pressures also affect the decision and strategies of institutions, which ultimately affects the performance of the employees [69].

Moreover, according to the study in [70], external drivers have a mediating effect on an organization's performance. Further, Huang and Yang [71] believes that coercive pressure has a role in the organization's employees' performance [72]. It means that coercive pressure plays a leading role in the development and performance of an organization and its employees. Furthermore, according to [73,74], innovation and development in the organization need to adapt according to the external environment for the organization's performance and its employees. Coercive pressure plays a vital role in the organization to adapt itself according to the current competitive environment.

Market pressure, regulatory pressure, and competitiveness pressure, are all factors that play a vital role in deploying the organization's business and enhancing the employees' performance [75]. According to [70], external environment pressure has a significant and moderating effect on the organization to compete in a competitive environment. It means that external institutional pressure plays a central role in enhancing organizational productivity. Moreover, coercive pressure improves an organization's performance directly and indirectly [76,77].

Further, Sahadev [78] explored how coercive pressure plays a moderator role between coordination and employees' performance. His findings expound that external institutional pressure plays a primary role in the isomorphism process of the organization. Further, Dubey et al. [74] also elaborated that external institutional pressures enhance coordination and collaboration between the institutions for better organization performance. Further, the institutional environment also affects the psychological safety of the employees [49,79].

Based on the previous research, it is considered that there exists a link between coercive pressures and employees' performance. Hence, this is considered as one of the main theoretical voids in the literature. Coercive pressure has a significant interaction between coordination, psychological safety, job security, and employee performance based on the discussed arguments. Based on the above discussion, the hypotheses are as follows:

**Hypothesis 4 (H4).** *Coercive pressure has a positive and significant relationship with employee performance.*

**Hypothesis 5 (H5).** *Coercive pressure plays the role of moderator between coordination and employees' performance.*

**Hypothesis 6 (H6).** *Coercive pressure strengthens the relationship between psychological safety and employees' performance.*

**Hypothesis 7 (H7).** *Coercive pressure has a strong relationship between job security and employees' performance.*

## 3. Materials and Methods

### 3.1. Data Collection and Participants

The target group of the study was HEIs' faculties and administration. A standard questionnaire was employed to collect the data from faculty members of HEIs.

Based on the simple random sampling technique, 26 HEIs were selected for this study. Three hundred questionnaires were distributed among the employees; 250 responses were recorded. The HEIs' faculty members were asked to record their responses by filling out the standard questionnaire. The participants responded against every variable's items based on the 5-point scale from 1 = strongly disagree to 5 = strongly agree. Based on the eligibility criteria and careful checking, we only included 235 responses for this study.

The higher education institutions' faculties list was downloaded from the higher education commission (HEC) website. Moreover, the HEIs' heads and registrar were also approached for the data collection.

### 3.2. Measurement

In the present study, the measurement items were adapted from existing literature and tested scales of employees' performance, coordination, psychological safety, and job security [46,64,72–76]. The strategy of inclusion and exclusion was adapted to make minor changes in the words and sentence structure. The details of the scale of study variables are given below and Appendix A.

#### 3.2.1. Employees' Performance

Items for employees' performance were adapted from [80]. The generated value of $\alpha$ for the adopted items was $\alpha = 0.83$.

#### 3.2.2. Coordination

The coordination scale was adapted from [81,82]. The items values were ($\alpha = 0.814$).

#### 3.2.3. Job Security

Job security was assessed through three items, adapted from Probst [83]. The scale explained item values of ($\alpha = 0.85$ and KMO = 0.717).

#### 3.2.4. Psychological Safety

Three items were used to assess psychological safety, adapted from [47,84] scale. The value of items were ($\alpha = 0.811$).

#### 3.2.5. Coercive Forces

The five items were adapted from [70] to assess the coercive forces. This scale explains ($\alpha = 0.956$) in the study.

#### 3.2.6. Common Method Bias (CMB)

A single source was used to collect data from the respondents. So, there may be the existence of a common bias issue in the study data set [85]. If the data has a variance greater than 50%, then it explains the CMB problem. Harman's single-factor approach was used to examine the CMB issue in the data. All of the items in this study data are divided into



five constructs. The first construct showed 40% of the total variance, which was less than 50% [86]. Therefore, there is no such common variance problem in the data. All the values of the VIF are less than the threshold value of 3.3; these results show that there is no such issue of CMB in our data [87].

## 4. Descriptive Statistics

Table 1 demonstrates the characteristics of the sample, which indicates that in the data collection process, most of the respondents were male (*n* = 235, 65.1%), and only (*n* = 235, 34.9%) were females. On the other side, (*n* = 235, 36%) respondents were professors (associate, assistant), and 36% were lecturers. In terms of age ratio, the number of samples aged 20–25 was 83, accounting for 35.2%; the number of samples aged 26–30 is 70, accounting for 29.8%; the number of samples aged 31–35 was 59, the proportion was 25.1%; for 34–40 years, respondents were 16 with a proportion of 6.8%, and for 41 and above age, participants were 12, and their proportion was 6.8%. In the sample of age, most of the respondents were between the ages of 20–25 and, 26–30 years, which elaborates that most of the respondents were young. In terms of the proportion of education, the number of respondents of master education holders was 188, accounting for 80%, and the number of respondents with Ph.D./other level degrees was 47, and the proportion is 20%. The majority of the respondents had a master's degree.

**Table 1.** Descriptive statistics.

| Sample Information | Types | Number of Samples | Percentage |
|---|---|---|---|
| Gender | Male | 153 | 65.1% |
| | Female | 82 | 34.9% |
| Designation | Lecturers | 150 | 64% |
| | Professors | 85 | 36% |
| Age | 20–25 years old | 83 | 35.3% |
| | 26–30 years old | 70 | 29.8% |
| | 31–35 years old | 59 | 25.1% |
| | 36–40 years old | 16 | 6.8% |
| | 41 years old and above | 12 | 3.0% |
| Education | Master/MS/M.Phil | 188 | 80% |
| | Ph.D. or other | 47 | 20% |

### 4.1. Measurement Model

In this study regression, four steps approach from Baron and Kenney (1986) was used to analyze the data. The reliability and validity of the scale were also checked [88]. The structural equation modeling technique in SmartPls was used to analyze the relationship between the data. This method is preferred because it estimated the multiple and interrelated dependence in a single analysis [88,89].

#### 4.1.1. The Confirmatory Factor Analysis

Confirmatory factor analysis (CFA) was used to check the model fitness test. According to this (see Table 2), all the requirements are full for a model fitness [89]. Results proved that our hypothesized five-factor model is fit (see Table 2), and values were within the range.

**Table 2.** Reliability and convert validity.

| Construct | Items | Loadings | Cronbach's Alpha | CR | AVE |
|---|---|---|---|---|---|
| Coordination | CO1 | 0.782 | 0.712 | 0.839 | 0.635 |
| | CO2 | 0.791 | | | |
| | CO3 | 0.816 | | | |
| Coercive Pressure | CP1 | 0.828 | 0.763 | 0.863 | 0.678 |
| | CP2 | 0.863 | | | |
| | CP3 | 0.778 | | | |
| Employees' Performance | EP1 | 0.825 | 0.84 | 0.893 | 0.676 |
| | EP2 | 0.858 | | | |
| | EP3 | 0.84 | | | |
| | Ep4 | 0.762 | | | |
| Job Security | JS1 | 0.848 | 0.665 | 0.81 | 0.588 |
| | JS2 | 0.748 | | | |
| | JS3 | 0.698 | | | |
| Psychological Safety | PS1 | 0.787 | 0.63 | 0.802 | 0.575 |
| | PS2 | 0.773 | | | |
| | PS3 | 0.714 | | | |

Note(s): All factor loadings are significant at $p < 0.001$; CR = Composite Reliability and AVE = Average Variance Extracted.

### 4.1.2. Validity and Reliability

In this study, discriminant and convergent validities were checked. Table 2 expounds the factor loading values and average variance extracted (AVE), which confirmed convergent validity. To analyze discriminant validity, we work according to the guidelines of the [88] technique by comparing the shared variance between the variables with the AVE of each variable, and the results confirmed discriminant validity AVE > shared variance. The linear regressions were used among the variables in the study. Before applying linear regression analysis to measure and validate a study variable's model fit, some preliminary assumptions of the regression analysis like normality, outliers, homogeneity, and collinearity were checked. All these assumptions were fulfilling the regression requirements (collinearity, normality, outliers, and homogeneity were checked). The VIF values of all the items of the variables are less than the threshold value of 3.3%. Further, Figure 2 explains a good measurement model for the study. All these things explain a good fit of the model.

Table 3 depicts that the square root of the AVE is higher than the intercorrelation among constructs which indicates a good discriminant validity [90]. In Table 3, the intercorrelation of the variables is less than the square root of AVE.

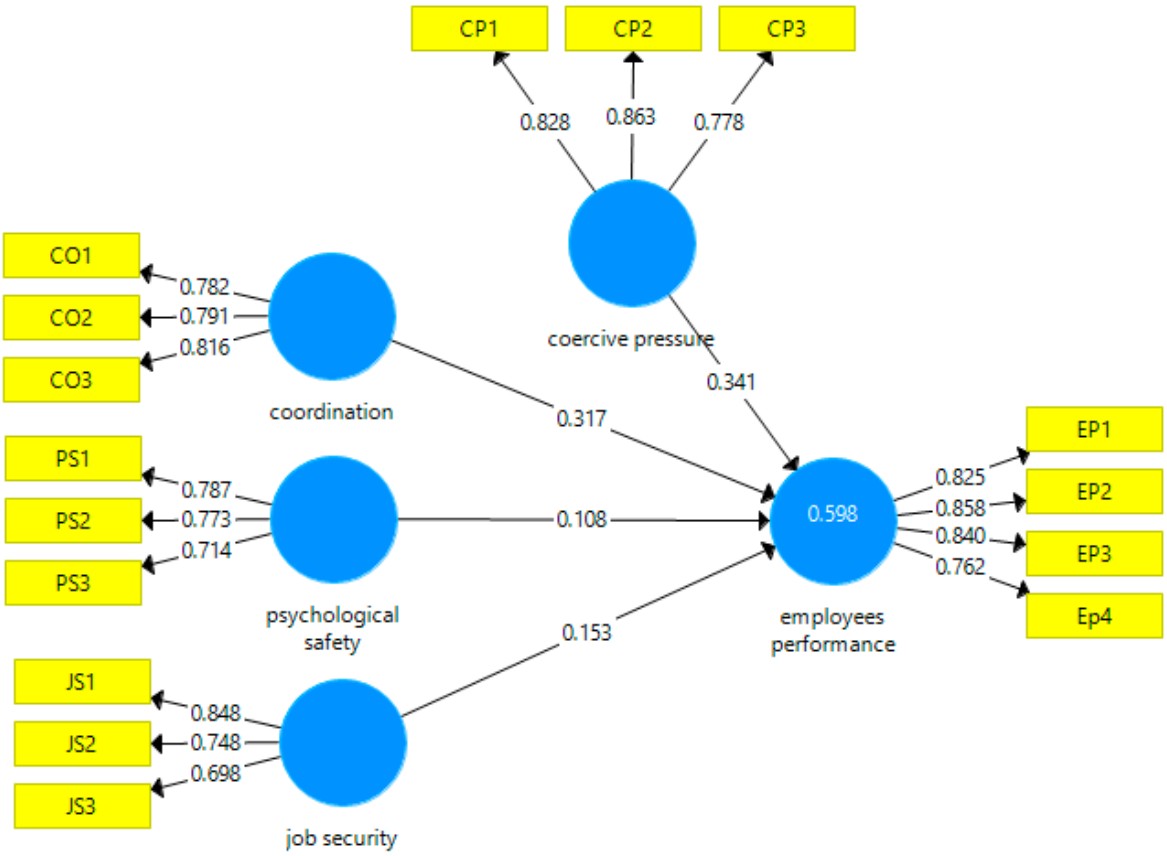

**Figure 2.** Showing the relationship between independent and dependent variables.

**Table 3.** Correlation between the constructs.

|  | Coercive Pressure | Coordination | Employees' Performance | Job Security | Psychological Safety |
|---|---|---|---|---|---|
| Coercive pressure | 0.824 |  |  |  |  |
| Coordination | 0.611 | **0.797** |  |  |  |
| Employees' performance | 0.677 | 0.685 | **0.822** |  |  |
| Job security | 0.469 | 0.427 | 0.502 | **0.767** |  |
| Psychological safety | 0.648 | 0.875 | 0.682 | 0.499 | **0.758** |

Figure 2 illustrates a positive and significant relationship between coordination, psychological safety, job security, coercive pressure, and employees' performance. Further, all the independent variables explain a variation of 59.8% in employees' performance.

*4.2. Testing of Hypotheses*

A regression analysis was carried out to test the direct relationship of the coordination, job security, psychological safety relationship with employees' performance, as well as the moderating effect of coercive pressure in the relationship between coordination, job security, psychological safety, and employees' performance. Figure 3 below showing the structural model detail. The results are as follows.

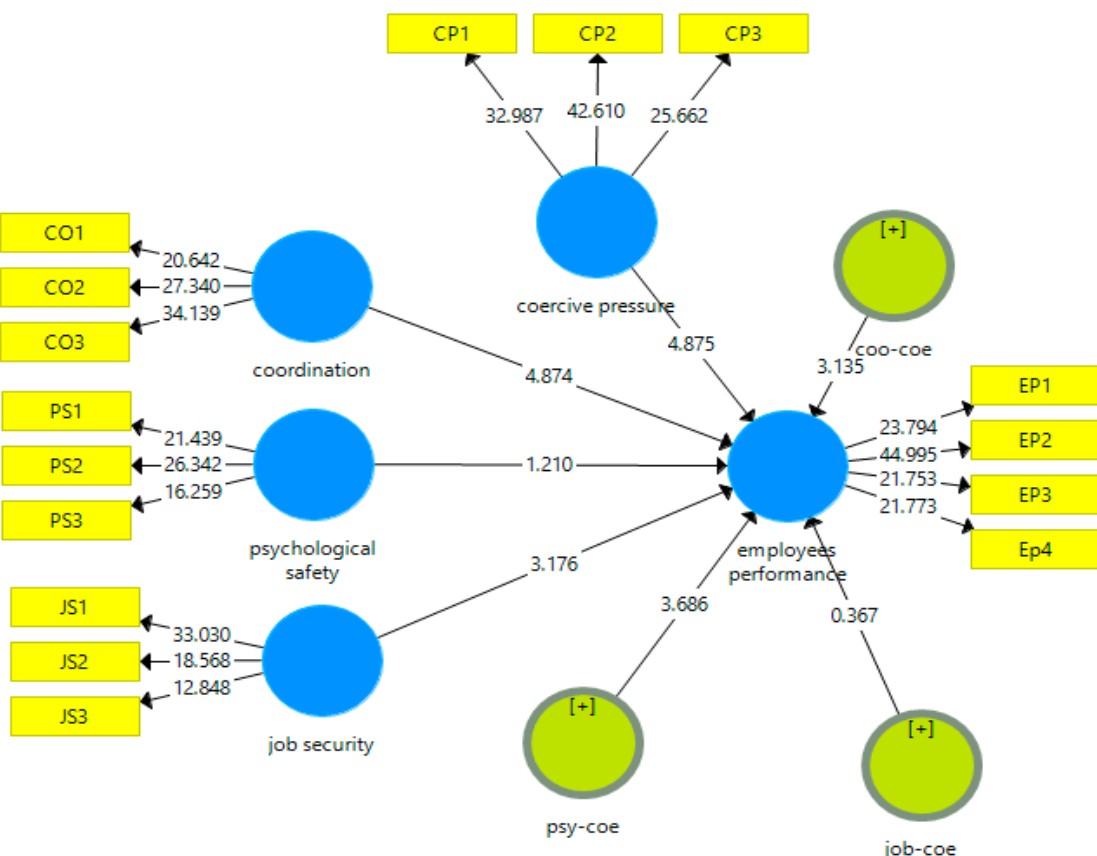

**Figure 3.** The relationship between independent variables, moderating variable and dependent variable.

### 4.2.1. Coordination

Table 4 illustrates that coordination positively and significantly influences employees' performance. Coercive pressure also has a positive relationship with employees' performance. Therefore, our study results supported Hypothesis **H1** and **H4.** Further, Table 4 findings and Figure 4 explain that coercive pressure is dampening the positive relationship of coercive pressure between coordination and employee performance. So, results do not support Hypothesis **H5.**

**Table 4.** Moderating effect of coercive pressure between coordination and employees' performance.

| | Original Sample (O) | Sample Mean (M) | Standard Deviation (STDEV) | T Statistics (｜O/STDEV｜) | *p* Values |
|---|---|---|---|---|---|
| coordination -> employees' performance | 0.473 | 0.473 | 0.097 | 4.874 | 0.000 |
| coercive pressure -> employees' performance | 0.29 | 0.29 | 0.059 | 4.875 | 0.000 |
| coordination-coercive pressure-> employees' performance | −0.335 | −0.325 | 0.107 | 3.135 | 0.002 |

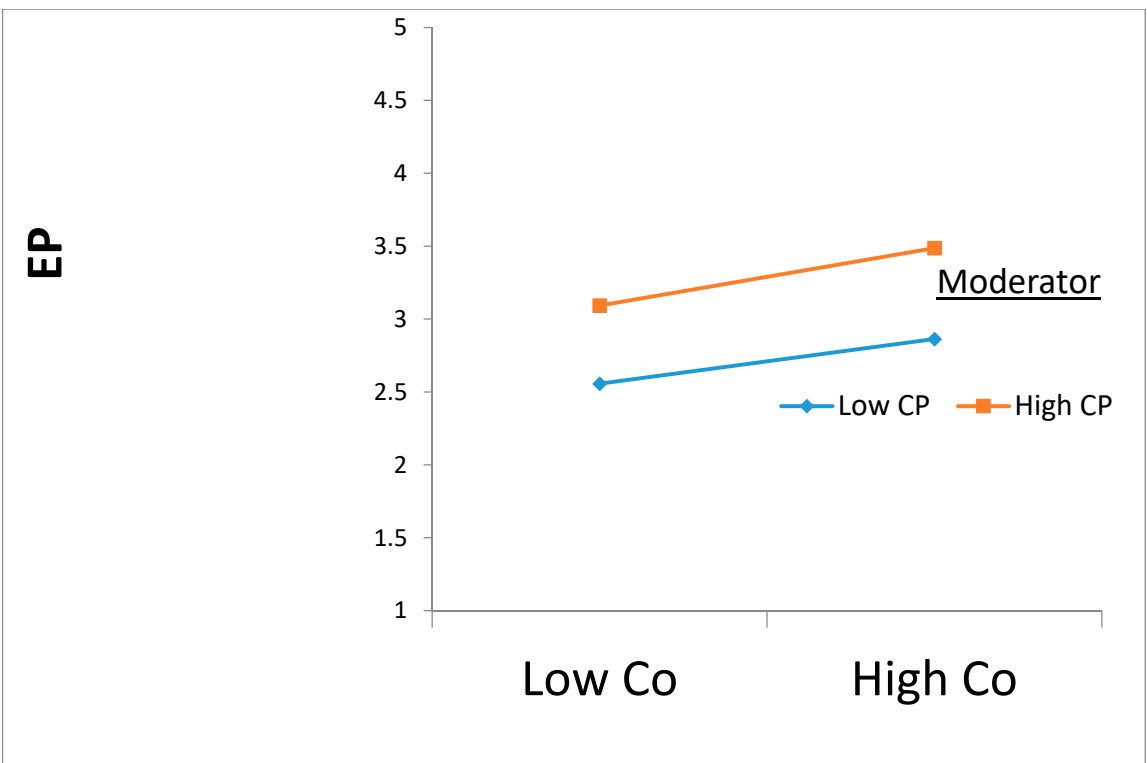

**Figure 4.** Coercive pressure dampens the positive relationship between coordination and employees' job performance.

### 4.2.2. Psychological Safety

Table 5 illustrates the psychological safety results (β = −0.133, *p* > 0.001), which show that there is a negative and insignificant relationship between psychological safety and employee' performance. So, Hypothesis **H2** is not supported. However, there is a positive and significant relationship between coercive pressure (β = 0.29 *p* < 0.001) and employees' performance. Therefore, Hypothesis **H4** is supported. Table 5 and Figure 5 show that coercive pressure has a dampening on the relationship between psychological safety and employees' job performance. So, Hypothesis **(H6)** is supported.

**Table 5.** Moderating effects of coercive pressure between psychological safety and employees' performance.

|  | Original Sample (O) | Sample Mean (M) | Standard Deviation (STDEV) | T Statistics (|O/STDEV|) | *p* Values |
|---|---|---|---|---|---|
| psychological safety -> employees' performance | −0.134 | −0.133 | 0.111 | 1.21 | 0.227 |
| coercive pressure -> employees' performance | 0.29 | 0.29 | 0.059 | 4.875 | 0.000 |
| Psychological safety-coercive pressure -> employees' performance | 0.418 | 0.4 | 0.113 | 3.686 | 0.000 |

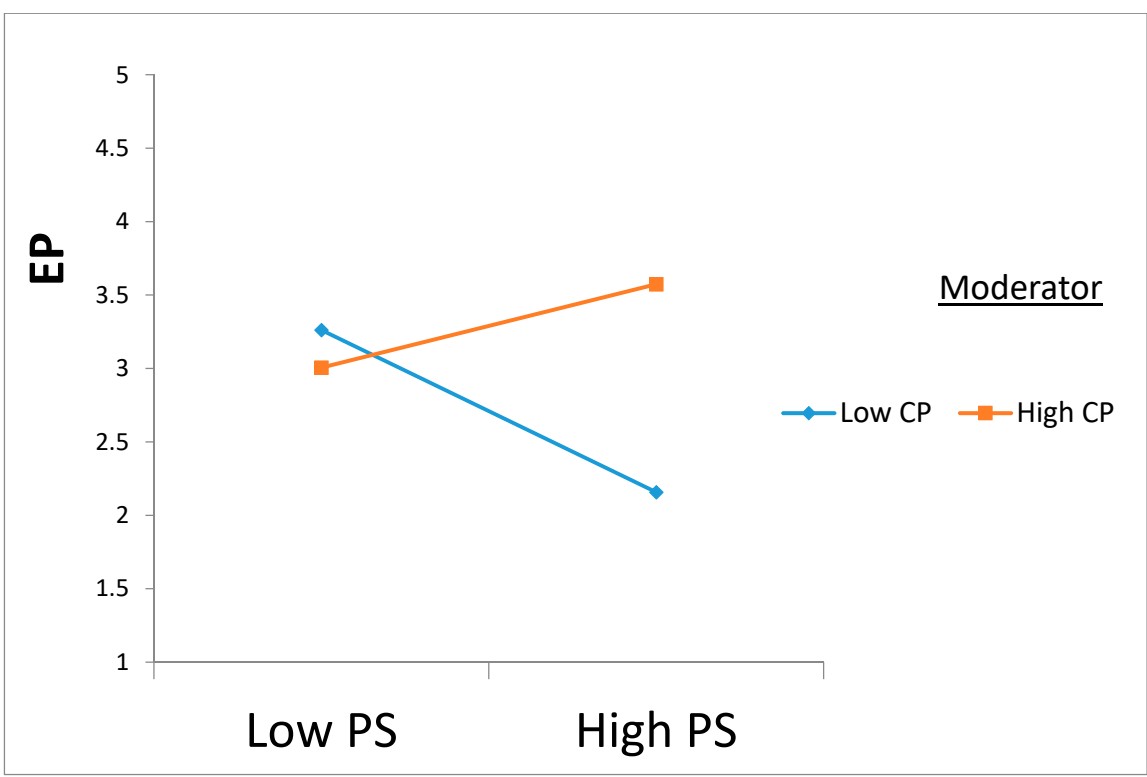

**Figure 5.** Coercive pressure dampens the relationship between psychological safety and employees' performance.

### 4.2.3. Job Security

Table 6 illustrates that in the job security ($\beta$ = 0.175 3, *p* < 0.001) results, there is a positive and significant relationship between job security and employees' performance. Further, coercive pressure has a positive and significant relationship to employees' performance. Hence Hypotheses **H3** and **H4** are supported. Further, Table 6 and Figure 6's results ($\beta$ = 0.055, *p* < 0.001) show that coercive pressure has a moderating role between job security and employees' job performance. Therefore, results support Hypothesis **(H7).**

**Table 6.** Moderating effect of coercive pressure between job security and employees' performance.

|  | Original Sample (O) | Sample Mean (M) | Standard Deviation (STDEV) | T Statistics (∣O/STDEV∣) | *p* Values |
|---|---|---|---|---|---|
| coercive pressure -> employees' performance | 0.29 | 0.29 | 0.059 | 4.875 | 0.000 |
| job security -> employees' performance | 0.174 | 0.175 | 0.055 | 3.176 | 0.002 |
| Job security-coercive pressure -> employees' performance | 0.02 | 0.022 | 0.055 | 0.367 | 0.001 |

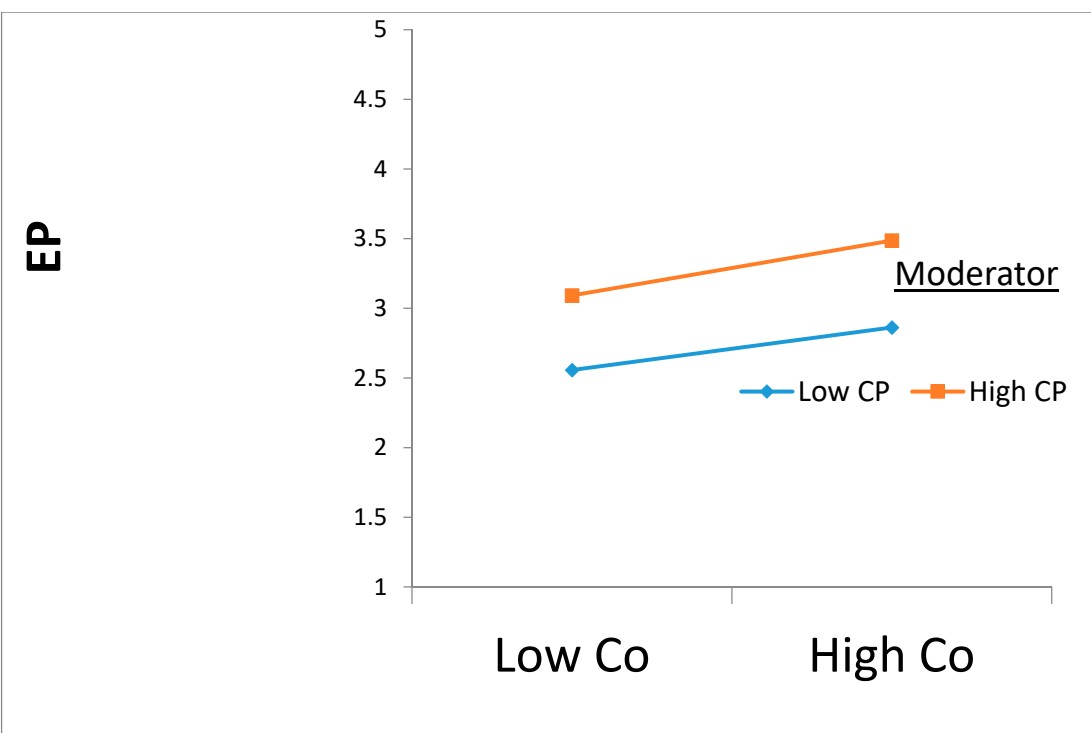

**Figure 6.** This figure shows that coercive pressure strengthens the relationship between job security and employees' performance.

## 5. Discussion

The study's primary purpose was to determine how coercive pressure moderates the relationship between coordination, job security, and psychological safety and how this affects employees' performance in Pakistan's private sector HEIs. These research questions were answered by testing the research framework used in this study by statistical analysis. Tables result in a display that show the coordination, psychological safety, and job security on employees' performance in HEIs in Pakistan.

The study's findings explain that coordination and employees' performance has a positive relationship with each other; these results are consistent with work [14,15]. Further, the moderating variable coercive pressure is dampening the positive relationship between coordination and employees' performance.

Moreover, we found that psychological safety has an insignificant relationship with the employees' performance. The variable results are not consistent with the findings of [91], in which they have the views that psychological safety has a positive and significant attachment to employees' performance. Further, study results report that coercive pressure has a partial moderating effect between psychological safety and employees' performance. The results contradict [92] in which coercive pressure has a moderating relationship between psychological safety and employees' performance, but these results are in line with [65,67]. These results demonstrated that institutional environments do not have a significant role in enhancing the performance of HEIs, but in the context of this study, these results support that institutional pressures affect performance of the employees. It shows that institutional pressure (coercive) has a role in enhancing the performance of the employees. However, a further study produced results that job security has a positive and significant predictor of employees' performance, and the study results corroborate with previous work of [93] in which they found that job security has a positive and significant effect on employees performance. Furthermore, this study's results elaborate that coercive pressure has a moderating role between job security and employee performance.

### 5.1. Theoretical Contribution

The current study contributes significantly to the literature on job security, coordination, psychological safety, coercive pressure, and employees' performance. This paper also indicates that coercive pressure has a moderating effect on coordination, psychological safety, job security, and employees' performance, and this relationship was unexplored in the literature of the studies done in the past. Previous studies only show us the psychological safety and employees' performance [26] with individual work performance [9], psychological safety with a mediating variable workplace bullying and harassment [11], coordination and employees' performance [14,15], relationship of coordination with job performance [15], and job stress and employees' performance [94]. There is no such relationship explored in the past studies between job security, coordination, psychological safety, and employees' performance with a mediating role of coercive pressure. Based on the above-mentioned relationship, further study was conducted in HIEs of Pakistan. This is a new contribution to the literature of institutional theory and employees' performance.

### 5.2. Practical Contribution

With the theoretical contribution in place, this study also has some practical contributions to the HEIs' main stakeholders. This suggests that they should focus on the effects of the institutional environment, particularly, coercive pressure in their institutions. Further, this research explains to the main actors that they should keenly observe the educational activities of HIEs for their better performance. Management should align the coercive pressure with employees' performance on a priority basis. Further administration should focus on coordination, psychological safety and job security on a priority basis because these factors have an impact on the employees' performance [95] and need to improve for the facilitation of the employees' performance. Furthermore, based on coordination, job security and psychological safety with a moderating effect of coercive pressure. performance of employees of HEIs can be enhanced [94]. This study suggests that HEIs' management thinks that coordination, job security, psychological safety, and institutional pressure are important for the best performance of their faculties.

### 5.3. Conclusions

This study purposes a theoretical model and hypothesizes the relationship between coordination, psychological safety, job security, and employee performance. Study results concluded a positive and significant relationship between coordination, job security, and employee performance. Moreover, there is a direct positive and significant association of job security with coercive pressure. Additionally, there exists a positive and significant relationship between coordination and coercive pressure.

Furthermore, there is a negative and insignificant relationship between psychological safety and employees' performance. Psychological safety also has a direct positive and significant relationship with coercive pressure. Likewise, coercive pressure has a direct link with the employees' performance. Finally, this study has tested the moderating role of coercive pressure between coordination, job security, psychological safety, and employee performance. Coercive pressure strengthens the relationship between coordination and job security positively.

Further, coercive pressure dampened the relationship between psychological safety and employee's performance. This study explained that employee performance varies due to the absence of job security, lack of coordination, and psychological safety. All the findings of the study explain that coercive pressure has a moderating role between coordination, job security, and psychological safety.

### 5.4. Limitations and Recommendations for Future Direction

Although this research study contributes to the literature on coordination, job security, psychological safety, institutional pressure (normative pressure), and employees' performance, it still has some limitations. First, in this study, only a cross-sectional approach was

used to collect the respondents' data. For in-depth analysis, a multi-method approach can be used. For instance, a semi-structured interview and disclosure analysis could be used. Secondly, our study examines both external and internal factors of HEIs that affect employees' productivity. Although only one institutional environment theory factor was taken in this study, in future studies, normative and mimetic pressures can also be considered as a mediator and moderating variables among coordination, job security, psychological safety, and employees' performance. Furthermore, the sample size of the study may be enlarged to generalize the results of the study for other higher education institutions.

Finally, our study's findings provide a footing for future research. Additionally, these study variables can be tested in other areas such as health care, the business sector, and small-medium enterprises to generalize the results. Along with these variables, employees' sustainable performance can be tested in future research. This research could also be extended to Pakistan's public sector HEIs for better investigations of the effects. Furthermore, the replication of the current study in multi-cultural perspectives across boundaries will develop more understanding of employees' performance, coordination, psychological safety, job security with institutional pressures (coercive, mimetic, normative). This study has implemented and validated the current model through the lens of institutional theory. In future avenues, learning organization theory may be used.

**Author Contributions:** Y.M.W., W.A. and H.L.Y.: These three have an equal contribution. They did the conceptualization, writing—original draft and methodology, formal analysis and validation and data curation. M.A., B.A. and Z.A. did the writing—review and editing. All authors have read and agreed to the published version of the manuscript.

**Funding:** The work is supported by Natural Science Foundation of China (No. 71640028, and 71362030), Key Laboratory of Educational Information for Nationalities (YNNU), and Ministry of Education.

**Institutional Review Board Statement:** Not applicable.

**Acknowledgments:** We would like to express special thanks and gratitude to reviewers and editors for their efforts in reviewing and correcting this paper manuscript.

**Conflicts of Interest:** The authors declare no conflict of interest.

## Appendix A.

*Appendix A.1. Employees' Performance [80]*

1. Are you aware of your work that you do being important for the institutions?
2. Do you learn new things while doing work in institutions?
3. Do you use your potential fully in your work?
4. Are you aware of the losses of the institutions if you do not work according to the requirements?

*Appendix A.2. Job Security [73]*

1. Are you satisfied with your present job or business in terms of job security?
2. Do you think that in the future you will lose your job?
3. Do you think that your job is guaranteed?

*Appendix A.3. Coordination [81,82]*

1. Do you prefer to work as a team with your colleagues?
2. Do you think that sharing information for better performance is important?
3. Do you think that your institutions' sharing of knowledge is strongly encouraged?

*Appendix A.4. Coercive Pressure [63]*

1. Institutions in our education sector that did not meet the legislated standards for learning faced a significant threat of legal prosecution from HEC.

2. Institutions in our education sector were aware of the fines and penalties potentially associated with environmentally irresponsible learning organization's behavior.

3. There were negative consequences for HEIs that failed to comply with the HEC environmental rules and regulations.

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
