# Peer review of "Impact of Coordination, Psychological Safety, and Job Security on Employees’ Performance: The Moderating Role of Coercive Pressure"

_sustainability, doi:10.3390/su13063175_

Round 1
Reviewer 1 Report
I recommend the following aspects to improve:
- The methodology used should be indicated, namely the modeling using structural equations as well as the reasons that led to its use.
- Limitations to the study and future recommendations should be presented at the end of the document.
Other aspects:
- On page 2, line 65, the question indicated should be b) and the rest of lines 66 and 67, c) and d).
- On page 4, line 159, the chapter numbering should be 2.3 (it is important to review the numbering).
- The figures should be captioned at the top as well as the tables keeping the same formatting
on page 13, line 318, the figure should be number 4. I consider that the figures and tables should have the source on which the author was based.
- Review the hypothesis format (line 139, 157,188. The size and alignment is different).
- Some references regarding the scales and coverage that support the items used should be presented as indicated in line 207.
Author Response
Reviewer 1 Comments and Suggestions for Authors I recommend the following aspects to improve:
• The methodology used should be indicated, namely the modeling using structural equations as well as the reasons that led to its use.
Author’s Response: Thanks for your grateful remarks; we have incorporated your suggestions in the paper. We use structural equation modelling in smartPLS software to analyze the data. This is the latest tool to analyze the data. This tool is recently used in the education sector. This tool is used in the complex model of the study. Further, This method is preferred because it estimate the multiple and interrelated dependence in a single analysis.
• Limitations to the study and future recommendations should be presented at the end of the document.
Author’s Response: we have incorporated some new limitations in the paper.
• On page 2, line 65, the question indicated should be b) and the rest of lines 66 and 67, c) and d).
Author’s Response: Thanks for your valuable comments. We have made changes according to your suggestion in numbering on page 2, lines 65,66 and 67 • On page 4, line 159, the chapter numbering should be 2.3 (it is important to review the numbering).
• The figures should be captioned at the top, as well as the tables keeping the same formatting
Author’s Response: Thanks for your valuable comments. We have incorporated page numbers according to your suggestions.
On page 13, line 318, the figure should be number 4. I consider that the figures and tables should have the source on which the author was based.
Author’s Response: Thanks for your valuable comments; we have incorporated table number according to sequence as number 7.
• Review the hypothesis format (line 139, 157,188. The size and alignment are different).
Author’s Response: Thanks for your valuable suggestions. We have incorporated your suggestion according to your comments.
• Some references regarding the scales and coverage that support the items used should be presented as indicated in line 207.
Author’s Response: Once again, thanks for your valuable remarks; we have incorporated your suggestions in the paper, which you can found in the track changes file.

Reviewer 2 Report
Thank you for a interesting paper, that can become a good contribution, but however, needs improvement. I liked the overall research design, but I have some topics that you need to address:
- I would like you to reflect on the empirical limitations of the study. I think that the empirical sample is not very big and it is only a certain group of people that you have investigated. You need to reflect on this limitation and what this does to the overall validity of your study. Indeed, it may be argued that this is only the case of this particular group of people in Pakistan and cannot be generalized. I think that you need to reflect much more on these limitations of your study.
- With regard to your theoretical concepts of psychological safety, I think the literature review should be improved of this concept. There must be much more recent literature and empirical studies from institutional theory that address this topic. Can you please update the literature review with regard to psychological safety.
- This also goes from your concept of coercive pressure. This concept is not clear in the paper. I really do not think that your conclusions are justified with regard to this concept. How can you say that coercive pressure improves the work performance? What is the justification of this? I think that when you compare to businesses and university organizations in Europe this is not the case so this conclusion needs to be further developed and justified with literature review and comparisons.
- Finally, some discussion of the general limitations of the study towards the end of the paper is needed.
- Good luck with your work!!
Author Response
Respected Vivien Lan
Assistant Editor
Sustainability
Subject: Revision and resubmission of sustainability-1116991 entitled " Impact of coordination, psychological safety, job security on employees’ performance: The moderating role of coercive pressure.”
Dated. February 27, 2021
Dear Assistance Editor,
Thank you for your letter and the opportunity to revise our paper on “Impact of coordination, psychological safety, job security on employees’ performance: The moderating role of coercive pressure.” The suggestions offered by the reviewers have been immensely helpful, and we also appreciate your insightful comments on revising the abstract and other aspects of the paper.
I have included the reviewer comments immediately after this letter and responded to them individually, indicating exactly how we addressed each concern or problem and describing the changes we have made. The revisions have been approved by all six authors and I have again been chosen as the corresponding author. The track changes are marked in the paper as you requested, and the revised manuscript is attached to this email message. We are very grateful for the reviews provided by the editors and each of the external reviewers of this manuscript. The comments are encouraging, and the reviewers appear to share our judgment that this study and its results are important. Please see below, in red, our detailed response to comments. All page numbers refer to the manuscript file with tracked changes.
Regarding more minor matters, we have now changed our spelling and phrasing. We apologize for neglecting that requirement in the author's instructions when we originally submitted the manuscript.
We hope the revised manuscript will better suit the International Journal of sustainability but are happy to consider further revisions, and we thank you for your continued interest in our research.
Sincerely,
The Authors
Reviewer 2
Comments and Suggestions for Authors
Thank you for an interesting paper that can become a good contribution, but however, needs improvement. I liked the overall research design, but I have some topics that you need to address:
- I would like you to reflect on the empirical limitations of the study. I think that the empirical sample is not very big, and it is only a certain group of people that you have investigated. You need to reflect on this limitation and what this does to the overall validity of your study. Indeed, it may be argued that this is only the case of this particular group of people in Pakistan and cannot be generalized. I think that you need to reflect much more on these limitations of your study.
Author’s Response: Thanks for your valuable comments and suggestions. Actually, in the COVID-19 have suffered a lot the education sectors. We tried to approach most of the faculty members, but the response rate remains very low due to this pandemic. We only selected Higher education Institutions faculty members as respondents. We have incorporated your comments in the limitations of the research.
- With regard to your theoretical concepts of psychological safety, I think the literature review should be improved on this concept. There must be much more recent literature and empirical studies from an institutional theory that addresses this topic. Can you please update the literature review with regard to psychological safety?
Author’s Response: Once again, thanks for your valuable comments. We have incorporated most of the new literature in the paper according to your suggestions. Regarding psychological safety literature, it is updated on page 3 from lines 121 to 128 and further from lines 134 and 144, 188, 189, and 190 representing some editing in literature.
- This also goes from your concept of coercive pressure. This concept is not clear in the paper. I really do not think that your conclusions are justified with regard to this concept. How can you say that coercive pressure improves work performance? What is the justification for this? When you compare to businesses and university organizations in Europe, this is not the case, so this conclusion needs to be further developed and justified with literature review and comparisons.
Author’s Response: Once again, for your valuable comments. We have incorporated your suggestions into the paper. Actually, in the context of Pakistan, the higher education commission (HEC) is a leading body of the education sector that monitors all the HEIs' activities. These rules and regulations are necessary to follow in Pakistan higher education institutions. The coercive pressure actually in the paper explains those rules and regulations, which are implemented by the HEC (main actor) to compel HEIs to work according to set standards of the HEC. If an organization is following these rules, it leads to the performance of the organization. On page 5, lines 169 to 176 are the new addition to the literature to further explain your comments/ suggestions.
- Finally, some discussion of the general limitations of the study towards the end of the paper is needed.
Author’s Response: We are grateful for your comments; we have incorporated general limitations at the end of the paper. In limitations on page 15 in lines 396, 398,399 and from lines 404 to 407, some changes are made.
Additional Comments: We have incorporated all the proofreading processes in the paper. Spelling and grammatical errors were checked and changes according to your valuable suggestions.

Round 2
Reviewer 1 Report
On page 3, there are sections 2.2.1 and 2.2.2 and 2.2.3, but there should be 2.2. The same goes on page 7, with section 4 missing. (It is necessary to revise the numbering of the sections)Author Response
Review1 comments
Comments and Suggestions for Authors
On page 3, there are sections 2.2.1 and 2.2.2 and 2.2.3, but there should be 2.2. The same goes on page 7, with section 4 missing. (It is necessary to revise the numbering of the sections)
Our Response. We have change numbering of the sections 2.2.1, 2.2.2 and 2.2.3 have changed as according to your suggestions in section 2, 3 , 4 and also in section 5. We highlight All the changes with red color.
Reviewer 2 Report
You have done minimum changes to comply with my comments. Thank you! I think the paper is now ready to be published. Congratulations!!
Author Response
Review2 comments
Comments and Suggestions for Authors
You have done minimum changes to comply with my comments. Thank you! I think the paper is now ready to be published. Congratulations!!
Our Response. Thanks for your valuable comments. We made as according to your suggestions. If there some other changes, we will come your comments. We incorporated most of the suggestions against your first review comments. Heartfelt thanks for you.